# IDTraffickers: An Authorship Attribution Dataset to link and connect Potential Human-Trafficking Operations on Text Escort Advertisements

**Vageesh Saxena**
Law & Tech Lab
Maastricht University
v.saxena@maastrichtuniversity.nl

**Benjamin Bashpole**
Bashpole Software, Inc.
bashpole@idtraffickers.com

**Gijs Van Dijck**
Law & Tech Lab
Maastricht University
gijs.vandijck@maastrichtuniversity.nl

**Gerasimos Spanakis**
Law & Tech Lab
Maastricht University
jerry.spanakis@maastrichtuniversity.nl

## Abstract

Human trafficking (HT) is a pervasive global issue affecting vulnerable individuals, violating their fundamental human rights. Investigations reveal that many HT cases are associated with online advertisements (ads), particularly in escort markets. Consequently, identifying and connecting HT vendors has become increasingly challenging for Law Enforcement Agencies (LEAs). To address this issue, we introduce IDTraffickers, an extensive dataset consisting of 87,595 text ads and 5,244 vendor labels to enable the verification and identification of potential HT vendors on online escort markets. To establish a benchmark for authorship identification, we train a DeCLUTR-small model, achieving a macro-F1 score of 0.8656 in a closed-set classification environment. Next, we leverage the style representations extracted from the trained classifier to conduct authorship verification, resulting in a mean r-precision score of 0.8852 in an open-set ranking environment. Finally, to encourage further research and ensure responsible data sharing, we plan to release IDTraffickers for the authorship attribution task to researchers under specific conditions, considering the sensitive nature of the data. We believe that the availability of our dataset and benchmarks will empower future researchers to utilize our findings, thereby facilitating the effective linkage of escort ads and the development of more robust approaches for identifying HT indicators [1].

## 1 Introduction

Human trafficking (HT) is a global crime that exploits vulnerable individuals for profit, affecting people of all ages and genders (EUROPOL, 2020; UNDOC, 2020). Sex trafficking, a form of HT, involves controlling victims through violence, threats,

deception, and debt bondage to force them into commercial sex (ILO, 2012). These operations occur in various locations such as massage businesses, brothels, strip clubs, and hotels (EUROPOL, 2020). Women and girls comprise a significant portion of HT victims, particularly in the commercial sex industry (ILO, 2012). Despite being advertised online, many victims have no control over the content of the advertisements (ads). Around 65% of HT victims are advertised online for escort services in the United States (POLARIS, 2020). However, the large number of online escort ads makes manual detection of HT cases infeasible, leading to numerous unidentified instances (POLARIS, 2018).

Researchers and law enforcement agencies (LEAs) rely on sex trafficking indicators (Ibanez and Suthers, 2014; Ibanez and Gazan, 2016; Lugo-Graulich and Meyer, 2021) to identify HT ads. However, these investigations require linking ads to individuals or trafficking rings, often using phone numbers or email addresses to connect them. Our research reveals that only 37% (202,439 out of 513,705) of collected ads have such contact information. Moreover, manual detection of HT cases is time-consuming and resource-intensive due to the high volume of online escort ads. To address this, researchers and LEAs are exploring automated systems leveraging data analysis (Keskin et al., 2021), knowledge graphs (Szekely et al., 2015; Kejriwal and Szekely, 2022), network theory (Ibanez and Suthers, 2014, 2016; Kejriwal and Kapoor, 2019; Kosmas et al., 2022), and machine learning (Dubrawski et al., 2015; Portnoff et al., 2017; Tong et al., 2017; Stylianou et al., 2017; Alvari et al., 2017; Shahrokh Esfahani et al., 2019; Wiriyakun and Kurutach, 2021; Wang et al., 2019). A recent literature review by (Dimas et al., 2022) highlights current trends on various research fronts for combating HT.

---

[1] Our code implementation is publicly available at https://github.com/maastrichtlawtech/IDTraffickers.git

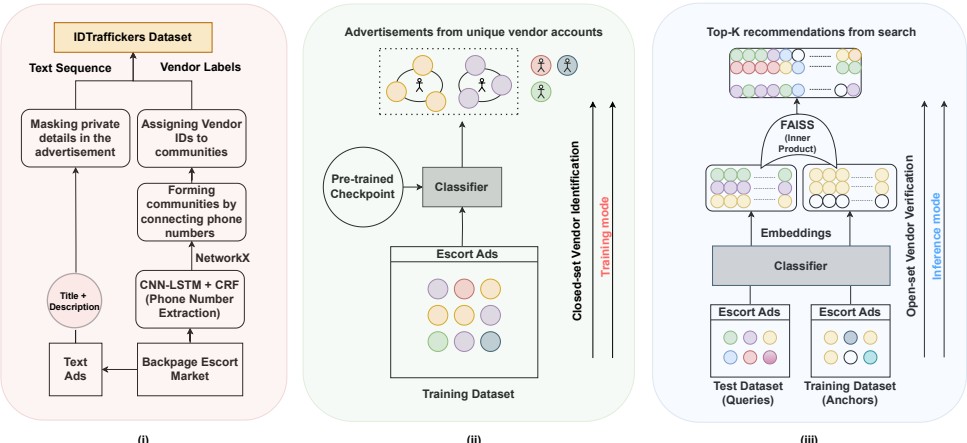

Figure 1: **(i) IDTraffickers:** Preparing authorship dataset from Backpage Escort Market, **(ii) Authorship Identification Task:** Identifying HT vendors in closed-set environment, **(iii) Authorship Verification Task:** Verifying HT vendors using advertisement similarity in open-set environment.

Although most of the abovementioned studies were conducted on online ads from the Backpage escort market, none analyzed authorship features to link and connect these trafficking operations. In the absence of phone numbers, email addresses, and private identifiers, such authorship techniques can become key to connecting vendor communities and analyzing the language, style, and content of escort ads. Therefore, as illustrated in Figure 1, this research focuses on bringing the following contributions to bridge the gap between authorship techniques and HT:

**(i) Authorship dataset:** Through this research, we release IDTraffickers, an authorship attribution dataset of 87.5K text ads collected between December 2015 and April 2016 from the United State's Backpage escort market [2]. Analyzing the language and content of these escort advertisements can provide crucial insights into the authorship traits and patterns associated with trafficking operations. Furthermore, by developing authorship attribution approaches on such a dataset, we can uncover recurring patterns and use them to link ads from potential HT communities, thereby bridging the gap in identifying and connecting individuals or groups

involved in trafficking operations.

**(ii) Authorship Benchmarks:** On escort markets, multiple vendors and communities often share a single account and post numerous ads. Moreover, some vendors create multiple accounts to avoid detection by LEA and expand their business. To address these challenges, we first establish an authorship identification (Green and Sheppard, 2013; Benzebouchi et al., 2019; Abbasi et al., 2022; Manolache et al., 2022; Saxena et al., 2023) benchmark through a closed-setting classification task (Vaze et al., 2022) (Figure 1(ii)). Given a specific text, the objective of the classifier is to predict the vendor that posted the advertisement. Furthermore, using the style representations from our trained classifier, we also establish an authorship verification (Stamatatos, 2016; Halvani et al., 2016; Benzebouchi et al., 2018; Manolache et al., 2022; Saxena et al., 2023) benchmark through an open-setting text-similarity-based ranking task (Figure 1(iii)). Given two ads, we compute the cosine similarity between the style representations to analyze the patterns in writing style and determine if they came from the same vendor.

## 2   Related Research

**Linking Escort Advertisements:** Identifying HT operations in escort ads is a challenging task that requires a multidisciplinary approach combining expertise in linguistics, data analysis, machine learning, and LEAs collaboration. Recent advancements in natural language processing (NLP) have

---

[2]Establishing ground truth for escort ads associated with HT is challenging as the undetected crimes are hard to identify. Even data based on arrest records may not encompass all undetected ads. To ensure the inclusion of HT ads, we analyzed Backpage escort website ads, which previous investigations have linked to HT ads (Callanan et al., 2017). Our primary objective is not to directly identify HT ads but to provide a tool that assists law enforcement in prioritizing resources by effectively detecting potential indicators of HT ads.

led researchers to identify indicators using entity linking approaches (Alvari et al., 2016; Nagpal et al., 2017a; Whitney et al., 2018; Alvari et al., 2017; Li et al., 2022) for automatic detection of HT operations. However, these indicators can only be studied within a cluster of ads linked with individual vendor accounts. Previous work by Chambers et al. (2019) proposed using neural networks to extract phone numbers to connect these escort ads. Nonetheless, our research demonstrates that only 37% of the ads in our dataset contained phone numbers. While clustering approaches (Lee et al., 2021; Vajiac et al., 2023a; Nair et al., 2022; Vajiac et al., 2023b) can assist in connecting near-duplicate ads, they fail to establish connections in paraphrased and distinct ads. Therefore, we focus our research on leveraging authorship techniques to analyze unique writing styles within escort ads and establish connections with individual vendors.

**Authorship Attribution in NLP:** In the past, research has established many machine-learning approaches to analyze text styles and link distinctive writing characteristics to specific authors. These approaches encompass TF-IDF-based clustering and classification techniques (Agarwal et al., 2019; İzzet Bozkurt et al., 2007), conventional convolutional neural networks (CNNs) (Rhodes, 2015; Shrestha et al., 2017), recurrent neural networks (RNNs) (Zhao et al., 2018; Jafariakinabad et al., 2019; Gupta et al., 2019), and contextualized transformers (Fabien et al., 2020a; Ordoñez et al., 2020; Uchendu et al., 2020; Barlas and Stamatatos, 2021). Moreover, researchers have recently demonstrated the effectiveness of contrastive learning approaches (Gao et al., 2022) for authorship tasks (Rivera-Soto et al., 2021; Ai et al., 2022). These advancements have led to applications in style representational approaches (Hay et al., 2020; Zhu and Jurgens, 2021; Wegmann et al., 2022), which currently represent the state-of-the-art (SOTA) for authorship tasks. Consequently, several datasets (Conneau and Kiela, 2018; Andrews and Bishop, 2019; Bevendorff et al., 2020, 2023) have been established to facilitate further research in this area.

**Authorship Attribution and Cybercrime:** Numerous authorship attribution studies have been successfully applied to the fields of forensic (Yang and Chow, 2014; Johansson and Isbister, 2019; Belvisi et al., 2020) and cybercrime investigations (Zheng et al., 2003; Rashid et al., 2013), spam de-

tection (Alazab et al., 2013; Jones et al., 2022), and linking vendor accounts on darknet markets (Ekambaranathan, 2018; Tai et al., 2019; Manolache et al., 2022; Saxena et al., 2023). However, to our knowledge, none of the existing studies focus on connecting vendors of HT through escort ads. In this research, we address this gap by introducing a novel dataset, IDTraffickers, which enables us to highlight the distinctions between language in existing authorship datasets and escort ads. Furthermore, we demonstrate the capabilities of authorship attribution approaches in establishing connections between escort advertisements and HT vendors using authorship verification and identification tasks.

## 3 Dataset

The data in this research is collected from online posted escort ads between December 2015 and April 2016 on the Backpage Market, a classified ads website similar to Craigslist on the surface web [3]. Although the market listing hosted everything from apartments to escorts, a report by Fichtner (2016) suggested that 90% of Backpage's revenue came from adult ads. Another report by Callanan et al. (2017) suggests that Backpage hosted escort listings concerned with the sex trafficking operations of women and children across 943 locations, 97 countries, and 17 languages. In this research, we accumulated 513,705 advertisements spread across 14 states and 41 cities in the United States.

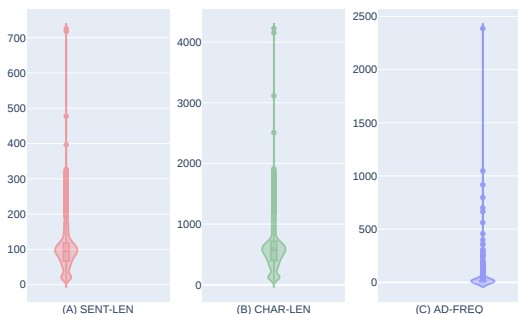

Figure 2: (A) Total number of tokens per ad (sent-len), (B) Total number of characters per ad, and (C) Number of ads per vendor (class-frequency) distributions.

[3]While we perform the pre-processing and restructuring of data for the authorship task, we would like to acknowledge Bashpole Software, Inc. for sharing the scraped data with us. To get access to our data, please visit our GitHub repository and follow the instructions provided in the link.

**Preprocessing:** First, we begin by merging the title and description of the text ads using the "[SEP]" token, as illustrated in Figure 1[i]. Figure 2(A) and figure 2(B) show that most ads in our dataset (approximately 99%) have a sentence length below 512 tokens and 2,000 characters. To generate ground truth, i.e., vendor labels, we employ the TJBatchExtractor Nagpal et al. (2017b) and CNN-LSTM-CRF classifier Chambers et al. (2019) to extract phone numbers from the ads. Subsequently, we utilize NetworkX Hagberg et al. (2008) to create vendor communities based on these phone numbers. Each community is assigned a label ID, which forms the vendor labels. For evaluation purposes, ads without phone numbers are discarded, resulting in a remaining dataset of 202,439 ads. Following the findings of Lee et al. (2021), which indicate that the average vendor of escort ads has 4-6 victims, we remove entries from vendors with fewer than five (average of 4-6) ads. The overall outcome of this process is a dataset comprising 87,595 unique ads and 5,244 vendor labels. Most of these vendors have an ad frequency of under 1,000 [4].

| Geography | Advertisements | Vendors |
|-----------|----------------|---------|
| **East** | 24,000 | 5,029 |
| **West** | 22,556 | 2,576 |
| **North** | 3,124 | 254 |
| **South** | 27,871 | 2,291 |
| **Central** | 21,124 | 2,928 |
| **Overall** | 87,595 | 5,244 |

Table 1: Total number of unique advertisements and vendors across US geography

After generating the vendor labels, we took measures to safeguard privacy by masking sensitive information within the ad descriptions. This included masking phone numbers, email addresses, age details, post ids, dates, and links to ensure that none of these details could be reverse-engineered, thereby minimizing the potential misuse of our dataset. Despite our efforts to extract escort names and location information using BERT-NER, RoBERTa-NER-based entity recognition techniques, and the approach described by (Li et al., 2022), we encountered a significant number of false positives. Unfortunately, our attempts to mask this information resulted in further noise in our data. Consequently, we decided to forgo this approach.

---

[4]For a more comprehensive understanding of our dataset, we encourage readers to find a detailed explanation in the datasheet attached in appendix A.4.

**Differences between Existing Authorship and IDTraffickers dataset:** To understand the differences between the existing authorship dataset and the IDTraffickers, we examine the part-of-speech (POS) and wikification (Szymanski and Naruszewicz, 2019) distributions between IDTraffickers, PAN2023 (Bevendorff et al., 2023), and the Reddit-Conversations dataset (Wegmann et al., 2022). The POS distribution is parsed through the RoBERTa-base spacy-transformers tagger (Montani et al., 2020), whereas the wikification is carried out using the Amazon ReFinED entity linker (Ayoola et al., 2022).

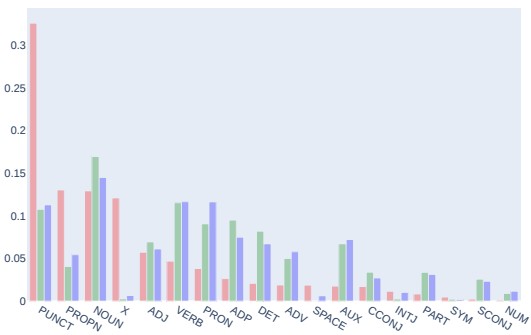

Figure 3: **POS-distribution:** Normalized POS-distribution for IDtraffickers, PAN2023, and Reddit-Conversations datasets.

Figure 3 presents a comparative analysis of the POS (part-of-speech) distributions among three datasets. The results reveal that the IDtraffickers dataset exhibits a higher frequency of punctuations, emojis, white spaces, proper nouns, and numbers than the other datasets. The punctuations, emojis, white spaces, and random characters represent approximately 47% of all POS tags in the IDtraffickers dataset. In contrast, these tags only account for 10.6% and 12.4% of all tags in the PAN2023 and Reddit conversation datasets, respectively. This discrepancy sheds light on the substantial noise within our dataset, highlighting the need for fine-tuning for domain adaptation.

In addition to examining POS distributions, we also investigate the wikifiability, or the presence of entities with corresponding Wikipedia mentions, on a per-advertisement basis. Figure 4 provides insights into the wikifiability across three datasets: IDTraffickers, PAN2023, and Reddit Con-

versational. Notably, the IDTraffickers dataset exhibits a higher level of wikifiability compared to the PAN2023 and Reddit Conversational datasets. However, a closer examination in figure 5 reveals that the majority of recognized entities in the ID-Traffickers dataset are primarily related to locations, escort names, or organizations. This observation aligns with the nature of the ads, as they often include information such as the posting's location, the escort's name, and nearby landmarks.

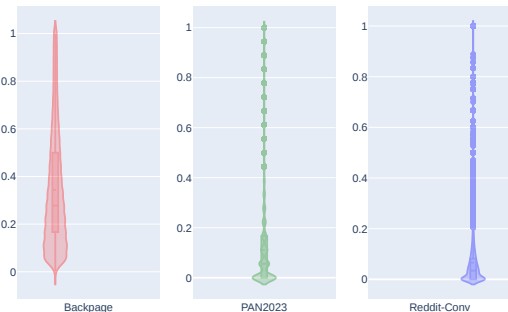

Figure 4: **Wikifiability:** No. of entities per advertisement with Wikipedia mentions in the IDtraffickers , PAN2023 , and Reddit-Conversations datasets.

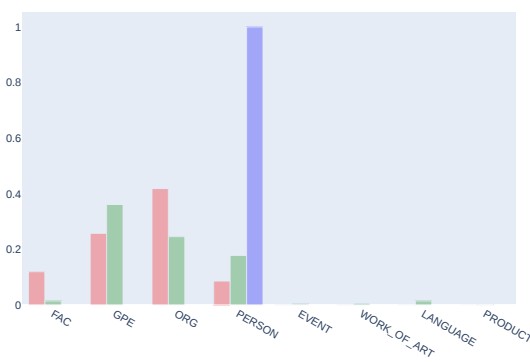

Figure 5: **Wiki-entities-distribution:** Extracted entities from the wikification of IDtraffickers , PAN2023 , and Reddit-Conversations datasets.

# 4 Experimental Setup

## 4.1 Authorship Identification: A Classification Task

Researchers have consistently demonstrated that the transformers-based contextualized models out-perform traditional stylometric approaches, statistical TF-IDF, and conventional RNNs and CNNs on authorship tasks (Kumar et al., 2020; Fabien et al., 2020b; Ai et al., 2022; Saxena et al., 2023). Hence, we establish our baselines through closed-set classification experiments using the distilled versions of BERT-cased (Sanh et al., 2020), RoBERTa-base (Liu et al., 2019), and GPT2 (Li et al., 2021), smaller versions of, AlBERTa (Lan et al., 2020) and DeBERTa-v3 (He et al., 2023), and architectures trained on contrastive objective such as MiniLM (Wang et al., 2020) and DeCLUTR (Giorgi et al., 2021). To account for domain differences, we also fine-tuned a RoBERTa-base language model (LM) on the IDTraffickers ads for the language task. Then, we extract the sentence representations from our trained LM and employ mean-pooling for the closed-set classification task [5]. In our research, we refer to this model as the LM-Classifier. Finally, we evaluated our results against a classifier trained on style representations (Wegmann et al., 2022), which currently represents the state-of-the-art (SOTA) in authorship tasks. We employed several metrics for evaluation, including balanced accuracy, micro-F1, weighted-F1, and macro-F1 scores. However, we emphasize the performance of our classifiers on the macro-F1 score due to the class imbalance (Figure 2(C)) in our dataset [6].

## 4.2 Authorship Verification: A Ranking Task

The closed-set classifier effectively identifies known vendors presented to it during training. However, it cannot handle inference for unknown vendors. Given the daily frequency of escort ads, it is impractical for law enforcement agencies (LEAs) to repeatedly train a network whenever a new vendor emerges. To address this limitation, we leverage the trained classifier to extract mean-pooled style representations from the ads and utilize FAISS (Johnson et al., 2019) for a similarity search. Specifically, we employ k-means clustering on the style representations. Our test set ads serve as query documents, while the training set ads act as index documents. By employing cosine similarity, we identify the K closest index documents for each query document. Since we treat the author-

---

[5]Please note that we experiment with mean, max, and mean-max pooling strategies for both author verification and identification tasks. However, the best results are obtained using the mean-pooling strategy

[6]The training setup and hyperparameter details are described in the appendix section A.1.

ship verification task as a ranking task, we evaluate the effectiveness of this similarity search operation using Precision@$K$, Recall@$K$, Mean Average Precision (MAP@$K$) (Pothula and Dhavachelvan, 2011; Jin et al., 2021), and average R-Precision scores (Beitzel et al., 2009) metrics.

## 5 Results

### 5.1 Authorship Identification Task

Table 3 showcase the performance of trained our classifier baselines, evaluated on the test IDTraffickers dataset.

| Models | Acc. | Micro-F1 | Weighted-F1 | Macro-F1 |
|---|---|---|---|---|
| **Distilled Models** | | | | |
| BERT | 0.9110 | 0.9147 | 0.9143 | 0.8467 |
| RoBERTa | 0.9199 | 0.9230 | 0.9229 | 0.8603 |
| GPT2 | 0.9132 | 0.9172 | 0.9166 | 0.8500 |
| **Smaller Models** | | | | |
| ALBERT | 0.7832 | 0.7891 | 0.7925 | 0.6596 |
| DeBERTa-v3 | 0.8703 | 0.8757 | 0.8756 | 0.7825 |
| T5 | 0.9157 | 0.9192 | 0.9190 | 0.8535 |
| **Contrastive Learning Models** | | | | |
| miniLM | 0.8888 | 0.8934 | 0.8935 | 0.8101 |
| **DeCLUTR** | **0.9230** | **0.9261** | **0.9259** | **0.8656** |
| Style-Emb | 0.8887 | 0.8936 | 0.8932 | 0.8112 |
| **HT Language Model** | | | | |
| LM-Classifier | 0.9294 | 0.9317 | 0.9316 | 0.8726 |

Table 3: Balanced Accuracy, Micro-F1, Weighted-F1, and Macro-F1 performances of the transformers-based classifiers on the author identification task.

As illustrated, the DeCLUTR-small classifier outperforms all other baselines for the authorship identification task. The success of the Style-Embedding model comes from its ability to latch onto content correlations. However, since escort ads have similar content types, the model struggles to effectively adapt to the noise in our data and distinguish between different writing styles. In contrast, DeCLUTR, trained to generate universal sentence representations, excels at capturing stylometric patterns and associating them with individual vendors. Next, we train the LM until convergence to achieve a perplexity of 6.22. When compared to the end-to-end DeCLUTR baseline, training a classifier over the trained representations of our LM only provides us with ∼1% increase in performance. We believe that the higher perplexity in the LM illustrates its inability to adapt to high noise in our data. Considering this minor improvement, we conclude that the additional training for our LM is not worthwhile. Consequently, we establish the DeCLUTR architecture as the benchmark for vendor identification on the IDTraffickers dataset.

### 5.2 Authorship Verification Task

Table 2 presents the open-set author verification results for the DeCLUTR and Style-Embedding models before and after being trained on the ID-Traffickers dataset. In this experiment, we conduct a similarity search on the training dataset for each query in the test dataset. The results show the average performance across all queries, along with the standard deviation. As can be observed, a substantial performance difference exists between the models before and after training, emphasizing the importance of our dataset. The zero-shot performance of the available checkpoints in red proves inconclusive for the author verification task.

The Precision@$K$ metric measures the model's

| K | @1 | @3 | @5 | @10 | @20 | @25 | @50 | @100 | @X |
|---|---|---|---|---|---|---|---|---|---|
| Precision@$K$ | | | | | | | | | |
| Style | 0.0442 ± 0.20 | 0.0410 ± 0.16 | 0.0391 ± 0.15 | 0.0366 ± 0.13 | 0.0329 ± 0.11 | 0.0319 ± 0.10 | 0.0270 ± 0.08 | 0.0227 ± 0.07 | - |
| DeCLUTR | 0.3198 ± 0.46 | 0.2883 ± 0.39 | 0.2671 ± 0.36 | 0.2278 ± 0.32 | 0.1837 ± 0.27 | 0.1693 ± 0.26 | 0.1277 ± 0.21 | 0.0893 ± 0.15 | - |
| Style | 0.9616 ± 0.19 | 0.9437 ± 0.19 | 0.9124 ± 0.21 | 0.8175 ± 0.27 | 0.6818 ± 0.33 | 0.6328 ± 0.35 | 0.4815 ± 0.36 | 0.3551 ± 0.36 | - |
| DeCLUTR | 0.9672 ± 0.17 | 0.9532 ± 0.17 | 0.9221 ± 0.19 | 0.8253 ± 0.26 | 0.6868 ± 0.33 | 0.6367 ± 0.34 | 0.4835 ± 0.36 | 0.3561 ± 0.36 | - |
| Recall@$K$ | | | | | | | | | |
| Style | 0.0023 ± 0.01 | 0.0063 ± 0.04 | 0.0091 ± 0.05 | 0.0146 ± 0.07 | 0.0233 ± 0.09 | 0.0269 ± 0.10 | 0.0394 ± 0.12 | 0.0580 ± 0.15 | - |
| DeCLUTR | 0.0242 ± 0.06 | 0.0567 ± 0.12 | 0.0792 ± 0.16 | 0.1136 ± 0.20 | 0.1539 ± 0.24 | 0.1676 ± 0.25 | 0.2122 ± 0.29 | 0.2590 ± 0.31 | - |
| Style | 0.0828 ± 0.09 | 0.2348 ± 0.24 | 0.3485 ± 0.32 | 0.5092 ± 0.37 | 0.6552 ± 0.37 | 0.6945 ± 0.36 | 0.7909 ± 0.32 | 0.8600 ± 0.27 | - |
| DeCLUTR | 0.0836 ± 0.09 | 0.2397 ± 0.25 | 0.3563 ± 0.32 | 0.5192 ± 0.37 | 0.6653 ± 0.37 | 0.7041 ± 0.36 | 0.7988 ± 0.32 | 0.8664 ± 0.27 | - |
| MAP@$K$ | | | | | | | | | |
| Style | 0.0442 ± 0.20 | 0.0562 ± 0.21 | 0.0598 ± 0.21 | 0.0640 ± 0.21 | 0.0673 ± 0.21 | 0.0681 ± 0.21 | 0.0700 ± 0.21 | 0.0712 ± 0.21 | - |
| DeCLUTR | 0.3198 ± 0.46 | 0.3587 ± 0.45 | 0.3681 ± 0.45 | 0.3750 ± 0.44 | 0.3794 ± 0.44 | 0.3803 ± 0.44 | 0.3823 ± 0.44 | 0.3833 ± 0.44 | - |
| Style | 0.9616 ± 0.19 | 0.9687 ± 0.16 | 0.9698 ± 0.15 | 0.9706 ± 0.15 | 0.9709 ± 0.14 | 0.9710 ± 0.14 | 0.9710 ± 0.14 | 0.9710 ± 0.14 | - |
| DeCLUTR | 0.9672 ± 0.17 | 0.9735 ± 0.14 | 0.9746 ± 0.14 | 0.9752 ± 0.13 | 0.9755 ± 0.13 | 0.9755 ± 0.13 | 0.9756 ± 0.13 | 0.9756 ± 0.13 | - |
| R-Precision@$X$ | | | | | | | | | |
| Style | - | - | - | - | - | - | - | - | 0.0199 ± 0.07 |
| DeCLUTR | - | - | - | - | - | - | - | - | 0.1641 ± 0.23 |
| Style | - | - | - | - | - | - | - | - | 0.8601 ± 0.22 |
| DeCLUTR | - | - | - | - | - | - | - | - | 0.8850 ± 0.20 |

Table 2: Precision@$K$, Recall@$K$, MAP@$K$, and R-Precision@$X$ scores for the DeCLUTR and Style-Embedding models before and after being trained on the IDTraffickers dataset

ability to identify relevant vendor ads within the top-$K$ predictions, whereas Recall@$K$ quantifies its capability to identify relevant recommended ads from the entire set of relevant ads in our training dataset. High precision at smaller $K$ values shows that our trained model retrieves relevant ads effectively within the top-$K$ retrieved items. While not all vendors in our dataset have a similar number of ads, we know they all have at least five ads. Therefore, we focus on precision performance for $K$ values less than or equal to 5. On the other hand, increasing Recall with higher $K$ values indicates that our trained model retrieves a large proportion of relevant ads for a query from the entire set of available relevant ads. Given that all vendors in our dataset have at least five ads each, we emphasize the recall performance of our trained model for $K$ values greater than or equal to 5. In addition to Precision@$K$ and Recall@$K$, we also evaluate the performance of our trained models using MAP@$K$, which considers the ordering of the retrieved items. It calculates the average precision across all relevant items within the top-$K$ positions, considering the precision at each position. As can be observed, our retrieval system prioritizes ads from the same vendor when presented with a query advertisement from vendor A. The high MAP scores achieved by our trained models for all K values indicate that the retrieved ads are effectively ranked, with the most relevant ones appearing at the top of the list.

Finally, we evaluate our trained models using the R-Precision metric, which focuses on precision when the number of retrieved ads matches the number of relevant ads ($X$) for a vendor. This metric disregards irrelevant items and solely concentrates on accurately retrieving relevant ones. The results show that the DeCLUTR-small models achieve an approximate 88% precision for retrieving relevant ads among the top-ranked items. In other words, our vendor verification setup effectively captures and presents the most relevant ads for a given query, achieving high precision at the rank equal to the number of relevant ads. Consequently, we establish the DeCLUTR-small model as the benchmark for the vendor verification task on the IDTraffickers dataset. However, there is significant fluctuation in standard deviation, indicating that precision varies depending on the vendor and ad frequency. Further investigation confirms this observation, with lower Precision@$K$, Recall@$K$, MAP@$K$, and R-Precision@$X$ values for vendors with fewer ads.

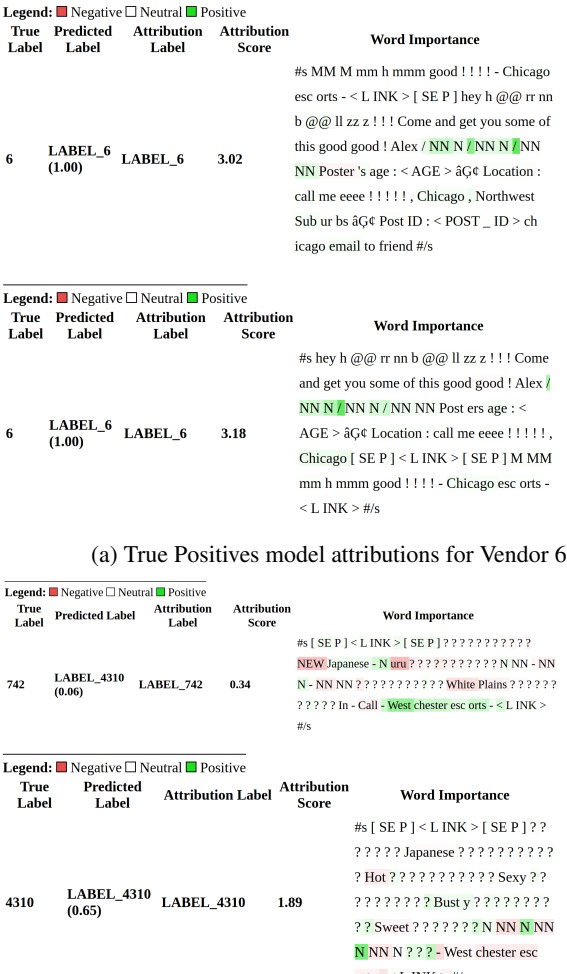

(a) True Positives model attributions for Vendor 6

(b) False Positives model attributions for Vendor 742

Figure 6: Model Explanations from the trained DeCLUTR-small classifier

## 5.3 Qualitative Analysis

To generate interpretable results, we use transformers-interpret (Pierse, 2021) build upon Captum (Kokhlikyan et al., 2020) to compute local word attributions in our text ads using the DeCLUTR-small classifier benchmark, indicating the contribution of each word to its respective vendor prediction. Figures 6a and 6b showcase True Positive and False Positive predictions from our trained classifier. Furthermore, for each query in the test dataset, we employ FAISS to identify the most similar ad in the training dataset. The figures depict positive (in green), zero, and negative (in red) word attribution scores for queries and their corresponding anchors associated with their vendor label.

Similar writing patterns and word attributions in the True Positive explanations confirm that the ads

are associated with the same vendor. This inference is supported by the consistent use of "@" and "/" preceding the digits of the masked phone number "/NN N / NN N/ NN NN." Conversely, the False Positive explanations shed light on instances where the model generated incorrect predictions, likely due to significant content and writing style similarities between vendors 4310 and 742. Both vendors frequently used a continuous sequence of "?" and mentioned Japanese services in their ads. Further examination reveals several instances where the classifier predicted the wrong vendor classes due to similar strong resemblances between the ads. This finding emphasizes the importance of carefully evaluating the quality of our classification labels. Note that vendor labels are established using the extracted phone numbers mentioned in the ads, which enabled us to connect ads and form vendor communities. In some cases, ads mentioned multiple phone numbers, aiding us in forming these connections. For others, the absence of this information led us to create a new vendor label. The hallucinations observed in our explanations result from this label assignment process, indicating the possibility of two vendors being the same entity.

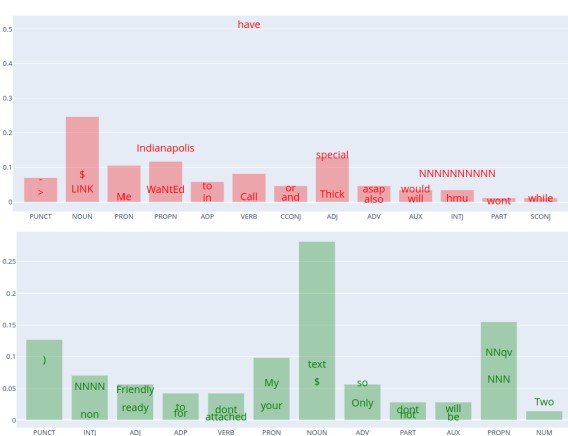

Figure 7: Word attribution collected over POS-distribution for ads of vendor 11178 and 11189 .

Inspired by Rethmeier et al. (2020), figure 7 employs global feature attributions to examine the discrepancies in writing styles between two specific vendors, namely vendors 11178 and 11189 . The analysis involves collecting word attributions and part-of-speech (POS) tags for all the advertisements from both vendors. The resulting bar plot presents the normalized POS density activated within the vendor ads and scatter points highlighting the two most attributed tokens for each POS

tag associated with the respective vendor [7]. The visualization clearly illustrates that both vendors employ distinct grammatical structures and word attributions in their advertisements, suggesting they are separate entities. This analysis enables law enforcement agencies (LEAs) to enhance their understanding of the connections between multiple vendors without solely relying on the ground truth.

## 6 Conclusion

In this research, we attempt to bridge the gap between connecting escort ads to potential HT vendors by introducing IDTraffickers, an authorship attribution dataset collected from the Backpage escort market advertisements (ads) within the United States geography. The dataset contains 87,595 ads with a text sequence of title and description and 5,244 vendor labels. Since these ads lack ground truth for the authorship task, we generate the labels by connecting the phone numbers. First, we establish a benchmark for the authorship identification task by training a DeCLUTR-small classifier with a macro-F1 of 0.8656 in a closed-set classification environment. Then, we utilize the style representations from the trained classifier to perform an open-set ranking task and establish a benchmark with an r-precision score of 0.8852 for the authorship verification task. Our experiments reveal a massive difference between the language in ID-Traffickers and existing authorship datasets. By performing the authorship identification task, we allow our classifier to adapt and benefit from this domain knowledge for the authorship verification task. Furthermore, we utilize the local and global feature attribution techniques to perform qualitative analysis on the trained classifier. Finally, our analysis reveals that most misclassifications in the trained classifier occur due to the possibility of multiple labels attributing to the same vendor. However, despite the lower performance, the classifier succeeds in generating style representations that allow us to identify these vendors through the ranking task. We believe that the availability of our dataset, benchmarks, and analyses will empower future researchers and LEAs to utilize the findings, aiding in the effective linkage of escort ads and developing more robust approaches to identifying HT indicators.

[7]Please note that we generate the plots using Plotly, which offers infinite zooming capabilities for any number of scatter points. However, we only display the two most attributed tokens for better clarity and visibility in the paper.

# 7 Limitations

**Assumption:** This research relies upon the classification task to adapt and benefit from domain knowledge. We assume each class label represents a different vendor in the classification process. However, our qualitative analysis reveals misclassification by the trained classifier due to heavy resemblance in writing style and content, indicating the possibility of multiple vendors being the same entity. While we cannot establish an absolute ground truth to validate our hypothesis, this presents a significant challenge. Nevertheless, we acknowledge that training a classifier with better-quality vendor labels would enhance the performance of our benchmarks.

**Dataset's Age:** Despite several active online escort platforms like Craiglist, YesBackpage, and Classified ads, we focus our research on Backpage ads spanning December 2015 to April 2016. This choice stems from documented human HT activities on the Backpage website (Callanan et al., 2017). We opted not to incorporate other active websites due to the absence of HT information in data from other escort websites. Please note that our research does not aim to track or assist currently ongoing human trafficking investigations. Instead, we aim to showcase the potential of authorship attribution techniques to aid law enforcement agencies in resource prioritization. Given this research objective, the age of the dataset holds smaller significance.

**Scalability:** In essence, different people write differently, and since the authorship task relies on identifying individual writing styles and perpetrators of human trafficking spans all racial, ethnic, and gender demographics (UNODC, 2023; Polaris, 2023); generalizing such approaches to vendors' existing years apart is not straightforward. Regarding using our approach on other escort websites, our methodology employs text ads to identify and establish connections among vendors associated with online escort ads. While the fundamental principles of our research can be theoretically extended to any active online escort website, the generalization capability of our approaches depends upon variations in writing styles, the use of chat-GPT-like automated systems, the presence of different languages, and data noise. In such instances, we recommend our readers improve the trained model through further fine-tuning, leveraging transfer learning techniques from the IDTraf-fickers dataset, and applying these to their specific data context.

**Larger Architectures:** Due to limited computational resources, we conducted our experiment using a distilled and small transformers-based architecture. It is worth noting that training the model on larger architectures has the potential to improve overall performance. Additionally, this research utilizes pre-trained representations to initialize our classifier architectures. However, incorporating supervised contrastive finetuning on our data can enhance the generation of stylometric representations, leading to better performance. Therefore, in the future, we plan to introduce a supervised contrastive pre-training baseline into our experiments.

**Zero-Shot Performance:** While we ensure not to use ads from the training dataset as queries for the ranking task, it is worth noting that the classifier was trained on the same dataset for the authorship identification task. To gain insights into the zero-shot capabilities of our trained representations, we intend to evaluate the authorship verification benchmark on unseen data. In order to achieve this, we plan to expand our data collection efforts, potentially sourcing data from other escort markets. This expansion will allow us to assess whether our model can generate stylometric representations that are universally applicable.

**Explainability:** This research employs local and global feature attribution techniques for qualitative analysis. However, it is important to address the limitations of these approaches (Das and Rad, 2020; Krishna et al., 2022). Local feature attribution techniques are susceptible to adversarial attacks and network sparsity. Moreover, they lack consideration for the broader context and dependencies in the data, and their explanations may not be entirely intuitive. Similarly, global feature attribution techniques suffer from a lack of granularity and contextual information. Both methods exhibit disagreements and significant inconsistencies in their explanations when applied to different XAI frameworks (Saxena et al., 2023). Therefore, in the future, we plan to develop more dependable explainability approaches that can help us better understand model behavior and ultimately foster trust amongst LEAs.

## 8 Broader Impact

**Data Protocols:** We collected our dataset from the Backpage Escort Markets, posted between December 2015 and April 2016 across various locations in the United States. In a related work (Krotov and Silva, 2018), the ethical implications of web scraping are characterized using seven guidelines. Adhering to these guidelines, we confirm that no explicit prohibitions are stated in terms of use policy on the Backpage website against data scraping. Furthermore, we intend to make our dataset available through the Dataverse data repository to mitigate any potentially illegal or fraudulent use of our dataset. The access to the data will be subject to specific conditions, including the requirement for researchers to sign a non-disclosure agreement (NDA) and data protection agreements. These agreements will prohibit sharing the data and its use for unethical commercial purposes. Considering that the data was collected before the seizure of the Backpage escort markets, we are confident that it does not pose any substantial harm to the website or its web server.

**Privacy Considerations and Potential Risks:** While we acknowledge the potential privacy concerns associated with the information within escort advertisements, we have extensive measures to mitigate these risks. These precautions include masking personal information, including phone numbers, email addresses, age details, post IDs, dates, and links. This masking is executed to prevent any possibility of reverse-engineering, thus mitigating the potential for misuse of our dataset. Unlike other encryption techniques, our masking process eliminates personal information. For instance, an email address like xzy@gmail.com is transformed into <EMAILID-23>, a phone number such as +00 00 0000 000 becomes +NN NN NNNN NNN, and a link like www.google.com is rendered as <LINK> within our dataset. Furthermore, as explained in the preprocessing section 3, we also experiment with various entity recognition techniques to try masking escort names and the posted locations mentioned in the advertisements. However, due to noise in our data, we encountered challenges in accurately masking these segments, resulting in false positive entity predictions. Nevertheless, considering that previous research indicates escorts often use pseudonyms in their advertisements (Carter et al., 2021; Lugo-Graulich) and no public records

of these advertisements exist after the seizure of Backpage Escort Markets in 2016, we find it unlikely that anyone can exploit the personal data in our advertisements to harm these individuals. Furthermore, given the legitimate interest of our research, our masking methodology aligns with the spirit of the General Data Protection Regulation (GDPR) article 6, the the lawfulness of processing, which acknowledges the importance of balancing privacy concerns with societal benefits. In this context, our research contributes to the betterment of society by facilitating studies aimed at addressing critical issues such as human trafficking and enhancing law enforcement efforts to reduce harm and save lives.

Finally, we intend to grant access to the ID-Traffickers dataset (link attached in our GitHub repository) via the Dataverse data portal. However, this access is subject to stringent restrictions. Researchers seeking access must undergo an application process, necessitating approvals and assurances. These assurances encompass non-disclosure and data protection agreements, ensuring responsible and secure use and release of the data.

**Legal Impact:** We cannot predict the specific impact of our research on the law enforcement process. Through this research, we aim to only assist Law Enforcement Agencies (LEAs) in comprehending vendor connections in online escort markets. Hence, we strongly recommend that LEAs and researchers not solely depend on our analysis as evidence for criminal prosecution. Instead, they should view our findings as tools to aid their investigations, not as direct evidence.

**Environmental Impact:** We conducted all our experiments on a private infrastructure, 100-SXM2-32GB (TDP of 300W), with a carbon efficiency of 0.432 $kgCO_2eq/kWh$. The training process for establishing all the baselines in our research took a cumulative 191 hours. Based on estimations using the Machine Learning Impact calculator presented in (Lacoste et al., 2019), the total estimated emissions for these experiments amount to 24.62 $kgCO_2eq$.

## 9 Acknowledgement

This research is supported by the Sector Plan Digital Legal Studies of the Dutch Ministry of Education, Culture, and Science, and Bashpole Softwares, Inc. Finally, the experiments were made

possible using the Data Science Research Infrastructure (DSRI) hosted at Maastricht University.

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

# A  Appendix

## A.1  Infrastructure & Schedule

**Split Ratio:** We perform data splitting into training, validation, and test datasets using a standard ratio of 0.75:0.05:0.20 for our experiments. To ensure reproducibility, we set the seed parameter to 1111 during this split.

**Training:** Our network training and evaluation are conducted on a Tesla V100 GPU with 32 GB of memory. To optimize the networks, we utilize the Adam optimizer with specific parameter configurations. The $\beta1$ and $\beta2$ values are set to 0.9 and 0.999, respectively. Additionally, we include an L2 weight decay of 0.01 in our optimization process. To determine the optimal learning rate, we experiment with values of 0.01, 0.001, and 0.0001. After analyzing the results, we find that the best

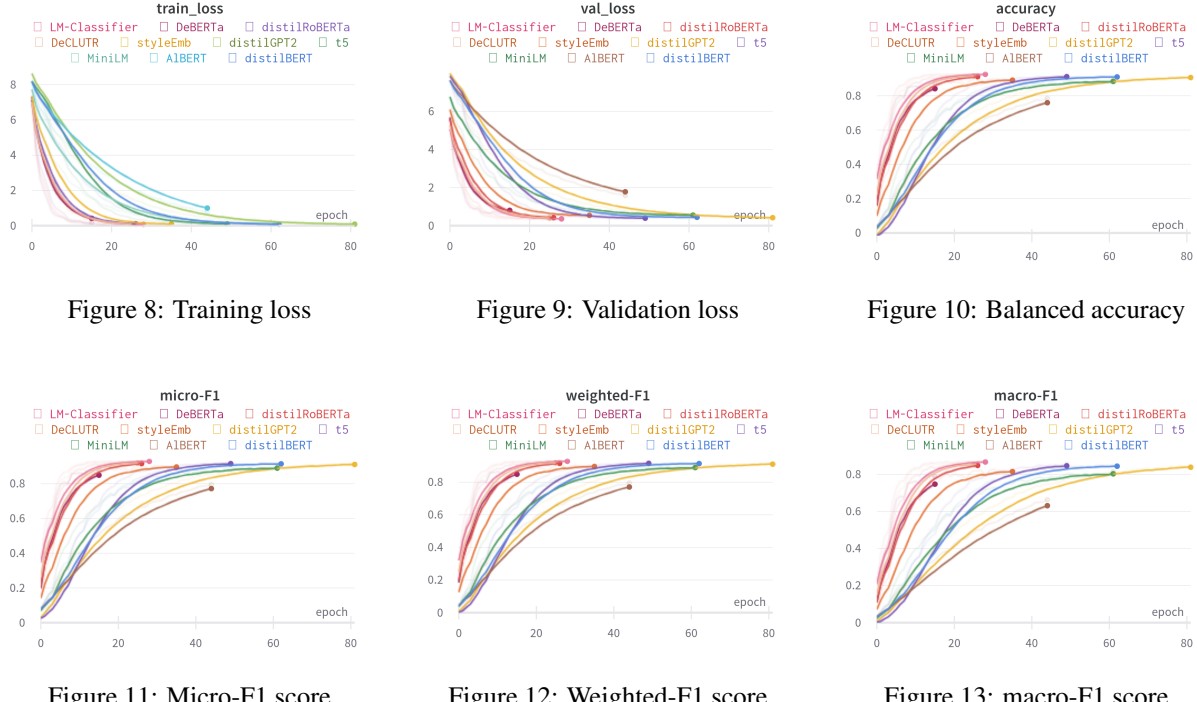

Figure 8: Training loss

Figure 9: Validation loss

Figure 10: Balanced accuracy

Figure 11: Micro-F1 score

Figure 12: Weighted-F1 score

Figure 13: macro-F1 score

Figure 14: Training loss, validation loss, and performance of trained classifiers on validation dataset.

performance is achieved when the learning rate is set to 0.001. Lastly, we employ a warm-up strategy for the initial 100 steps, followed by a linear decay.

**Architectures & Hyperparameters:** Considering the computational resources at our disposal, we initialize our transformer using pre-trained model checkpoints of DistilBERT-base-cased, DistilRoBERTa-base, DistilGPT2, all-MiniLM-L6-v2, AlBERT-small, DeBERTa-v3-small, T5-small, DeCLUTR-small, and Style-Embedding architectures. To perform the classification, we add a sequence classification head on top of each of these architectures. All models are trained until convergence, employing a batch size of 32 and a mini-batch of 4. We conducted experiments with lower batch sizes of 8, 16, and 20 but ultimately found that a batch size of 32 yielded the best results. Please note that 32 is the largest batch size we can compute without encountering memory issues during training.

All the experiments in our research are implemented in python3.9 (Van Rossum and Drake Jr, 1995) using Sklearn (Pedregosa et al., 2011), PyTorch (Paszke et al., 2019), Hugging-face (Wolf et al., 2020), and Lightning2.0 (Falcon and The PyTorch Lightning team, 2019) frameworks.

**Computational Details:** Tables 4 presents details about the number of trainable parameters, training time, and epoch of convergence for all the trained classifiers in our experiments. Please note that while training LM-Classifier on the author identification task takes the least time, training the language model on our dataset takes 65 epochs and additional 2 days, 7 hours, and 20 mins.

Figure 14 contains other training details, such as training and validation loss. Finally, we present the balanced accuracy, micro-F1, weighted-F1, and macro-F1 scores on the validation set for reproducibility purposes. We generate these results by exporting Wandb (Biewald, 2020) reports.

| Models | Param | Time | Epoch |
|---|---|---|---|
| **DistilBERT** | 66M | 25:44:00 | 63 |
| **DistilRoBERTa** | 82M | 11:13:05 | 27 |
| **DistilGPT2** | 82M | 25:45:00 | 82 |
| **all-miniLM-L6** | 22M | 21:50:04 | 62 |
| **ALBERT-small-v2** | 11.6 | 16:08:52 | 44 |
| **DeBERTa-v3-small** | 44M | 22:13:50 | 16 |
| **T5-small** | 35M | 19:03:35 | 50 |
| **DeCLUTR-small** | 86M | 05:53:33 | 26 |
| **Style-Embedding** | 128M | 18:43:06 | 36 |
| **LM-Classifier** | 86M | 04:47:10 | 19 |

Table 4: Total number of trainable parameters, training time, and convergence epoch for the trained classifiers.

**Random Initialization:** Due to limited resources, we only examine the effects of different initializations on our model's performance for the established DeCLUTR-small benchmark. Table 5 displays the mean and standard deviation in the model's performance against balanced accuracy, Micro-F1, Weighted-F1, and Macro-F1 scores. The results indicate minimal to no effects on these scores across different initializations.

| Seed | Acc. | Micro-F1 | Weighted-F1 | Macro-F1 |
|------|--------|----------|-------------|----------|
| 100  | 0.9256 | 0.9285 | 0.9286 | 0.8694 |
| 500  | 0.9243 | 0.9283 | 0.9286 | 0.8677 |
| 1000 | 0.9172 | 0.9205 | 0.9204 | 0.8559 |
| 1111 | 0.9230 | 0.9261 | 0.9259 | 0.8656 |
| 5000 | 0.9197 | 0.9229 | 0.9229 | 0.8598 |
| Mean | 0.9219 | 0.9252 | 0.9252 | 0.8636 |
| Std. | 0.0030 | 0.0031 | 0.0032 | 0.0050 |

Table 5: Influence of different initialization on DeCLUTR-small classifier's performance

## A.2 Language Distribution of Dataset

Our analysis reveals that approximately 99.32% of our dataset's vocabulary is English. Following that, in descending order, we find Spanish, Chinese, Japanese, Sotho, French, Bokmal, Dutch, Tswana, Nynorsk, Swedish, Latin, Basque, Swahili, Malay, Xhosa, and Welsh making up the remaining portion. Consequently, we did not explore the use of multilingual models in our research. Given that only a small fraction of our dataset's vocabulary lies outside English, we anticipate that employing multilingual models would not significantly improve performance. These statistics are obtained using the lingua language detection python package.

## A.3 Drift in vendor writing styles

The absence of ground truth poses challenges in collecting ads from authors spanning multiple years to comment on the stylometric drift. Our data collection specifically covers escort advertisements between December 2015 and April 2016, during which we observed no significant shifts in authors' writing styles. Moreover, given the brevity of this period, it is challenging for us to ascertain the stability of escort vendors. Notably, establishing the business age for potential human trafficking vendors is intricate, mainly because not all such vendors are apprehended by law enforcement. However, research indicates that more organized traffickers exploit victims extensively over extended timeframes (Lugo-Graulich and Meyer, 2021).

That said, we see a drift among vendor locations within our dataset. Around 75 vendors in our dataset post ads between east-west, 6 between east-north, 107 between east-south, 284 between east-central, 7 between west-north, 53 between west-south, 34 between west-central, 7 between north-south, 8 between north-central, and south-central American geographies. Our analysis further reveals that 7 vendors post their advertisements across all 41 states of the United States from which our dataset was collected.

## A.4 Datasheet

### A.4.1 Motivation

**For what purpose was the dataset created? Was there a specific task in mind? Was there a specific gap that needed to be filled? Please provide a description.** Law Enforcement Agencies (LEAs) and researchers currently use sex trafficking indicators to distinguish between online escort advertisements and Human Trafficking (HT) cases. These investigations involve connecting ads to specific individuals or trafficking networks, often by examining phone numbers or email addresses. However, existing methods for linking these escort ads have limitations. Therefore, we release IDTraffickers, an authorship attribution dataset, to identify and link potential HT vendors by analyzing and linking distinctive writing characteristics.

### A.4.2 Composition

**What do the instances that comprise the dataset represent (e.g., documents, photos, people, countries)? Are there multiple types of instances (e.g., movies, users, and ratings; people and interactions between them; nodes and edges)? Please provide a description.** The instances in the dataset comprise raw text sequences obtained by merging the title and description of the escort ads using the [SEP] token.

**How many instances are there in total (of each type, if appropriate)? What data does each instance consist of? "Raw" data (e.g., unprocessed text or images)or features? Is there a label or target associated with each instance?** In total, we collected 87,595 ads connected to 5,244 vendors. The dataset is released as a pandas dataframe in a .csv file with "TEXT" and "VENDOR" being the two columns. The "TEXT" column contains the input sequence of string format, whereas the "VENDOR" column contains class labels in int

format.

**Does the dataset contain all possible instances or is it a sample (not necessarily random) of instances from a larger set? If the dataset is a sample, then what is the larger set? Is the sample representative of the larger set (e.g., geographic coverage)? If so, please describe how this representativeness was validated/verified. If it is not representative of the larger set, please describe why not (e.g., to cover a more diverse range of instances, because instances were withheld or unavailable).** The IDTraffickers dataset comprises advertisements from the Backpage escort market, spanning locations across 14 states and 41 cities, and the United States. Figure 15 demonstrates the density of unique advertisements collected across American states. Initially, we collected a total of 513,705 ads. However, we filtered out ads that did not include phone numbers, as these phone numbers serve as the ground truth for our dataset. Consequently, our dataset consists of 202,439 ads that meet this criterion.

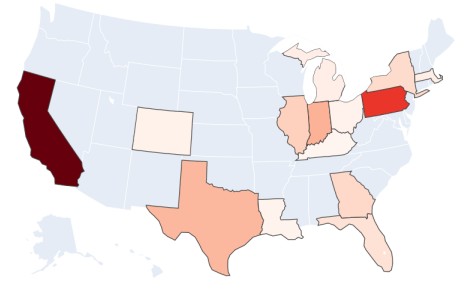

Figure 15: Density of unique advertisements collected across American states

**Is any information missing from individual instances? If so, please provide a description, explaining why this information is missing (e.g., because it was unavailable). This does not include intentionally removed information, but might include, e.g., redacted text.** No

**Are relationships between individual instances made explicit (e.g., users' movie ratings, social network links)? If so, please describe how these relationships are made explicit.** To generate ground truth labels for authorship task in our dataset, we employ the TJBatchExtractor Nagpal et al. (2017b) and CNN-LSTM-CRF classifier Chambers et al. (2019) to extract phone numbers from the ads. Subsequently, we utilize NetworkX Hagberg et al. (2008) to create vendor communities

based on these phone numbers. Each community is assigned a label ID, which forms the vendor labels.

**Are there recommended data splits (e.g., training, development/validation, testing)? If so, please provide a description of these splits, explaining the rationale behind them.** We perform our data splitting into training, validation, and test datasets using a standard ratio of 0.75:0.05:0.20 for our experiments.

**Are there any errors, sources of noise, or redundancies in the dataset? If so, please provide a description.** Our analysis in section 5.3 reveals the possibility of two or more vendor labels associated with the same entity. However, in the absence of absolute ground truth, we cannot make any conclusive statements. Figure 3 demonstrates the presence of significant noise within the data, such as punctuations, emojis, white spaces, and random characters deliberately inserted by escort vendors to evade automated systems.

**Is the dataset self-contained, or does it link to or otherwise rely on external resources (e.g., websites, tweets, other datasets)? If it links to or relies on external resources, a) are there guarantees that they will exist, and remain constant, over time; b) are there official archival versions of the complete dataset (i.e., including the external resources as they existed at the time the dataset was created); c) are there any restrictions (e.g., licenses, fees) associated with any of the external resources that might apply to a dataset consumer? Please provide descriptions of all external resources and any restrictions associated with them, as well as links or other access points, as appropriate** No. The dataset is self-contained.

**Does the dataset contain data that might be considered confidential (e.g., data that is protected by legal privilege or by doctor–patient confidentiality, data that includes the content of individuals' nonpublic communications)? If so, please provide a description.** While we acknowledge the potential privacy concerns associated with the information within escort advertisements, we have taken measures to mitigate these risks. Specifically, we have masked sensitive details such as phone numbers, email addresses, age information, post IDs, dates, and links within the advertisements. All the digits of phone numbers in our ads are replaced

by the letter "N." Similarly, we replace all the email addresses with $< EMAIL\_ID >$, post IDs with $POST\_ID : NNNNN$ where N represents a digit of a number, dates with $< DATES >$, and links with $< LINK >$. Since our dataset contains only text data, all the escort images are discarded. Furthermore, as explained in preprocessing section 3, we also experiment with various entity recognition techniques to try masking escort names and the posted locations mentioned in the advertisements. However, due to noise in our data, we encountered challenges in accurately masking these segments, resulting in false positive entity predictions. Nevertheless, considering that previous research indicates escorts often use pseudonyms in their advertisements (Carter et al., 2021; Lugo-Graulich) and no public records of these advertisements exist after the seizure of Backpage Escort Markets in 2016, we find it unlikely that anyone can exploit the personal data in our advertisements to harm these individuals.

**Does the dataset contain data that, if viewed directly, might be offensive, insulting, threatening, or might otherwise cause anxiety? If so, please describe why.** The dataset comprises text from escort advertisements that contains sexual descriptions of the services.

**Does the dataset identify any subpopulations (e.g., by age, gender)? If so, please describe how these subpopulations are identified and provide a description of their respective distributions within the dataset.** We mask all the age details of escorts in our dataset. However, the advertisements still contain a description of the escort ethnicities.

**Is it possible to identify individuals (i.e., one or more natural persons), either directly or indirectly (i.e., in combination with other data) from the dataset? If so, please describe how.** While it is not impossible for individuals to find connections from our dataset, we have taken extensive measures to minimize that risk by masking private identifiers in the ads. The dataset consists of ads from the Backpage escort markets collected between December 2015 and April 2016. Since the seizure of the website, no public records exist for these ads.

**Does the dataset contain data that might be considered sensitive in any way (e.g., data that reveals race or ethnic origins, sexual orientations,**

**religious beliefs, political opinions or union memberships, or locations; financial or health data; biometric or genetic data; forms of government identification, such as social security numbers; criminal history)? If so, please provide a description.** Despite our masking measures, our efforts to extract escort names, ads location, ethnicities, and sexual orientations failed due to the noise in our data. These details are mentioned in the text description of the ads.

### A.4.3 Collection Process

**How was the data associated with each instance acquired? Was the data directly observable (e.g., raw text, movie ratings), reported by subjects (e.g., survey responses), or indirectly inferred/derived from other data (e.g., part-of-speech tags, model-based guesses for age or language)? If the data was reported by subjects or indirectly inferred/derived from other data, was the data validated/verified? If so, please describe how.** To generate vendor labels in our dataset, we employ the TJBatchExtractor Nagpal et al. (2017b) and CNN-LSTM-CRF classifier Chambers et al. (2019) to extract phone numbers from the ads. Subsequently, we employ NetworkX Hagberg et al. (2008) to create vendor communities based on these phone numbers. Each community is assigned a label ID, which forms the vendor labels. Given the large size of our dataset and the abundance of phone numbers in the advertisements, manual validation of each extracted phone number is infeasible. However, we address this challenge by training the CNN-LSTM-CRF classifier using five random initializations to assess the robustness of the predictions. To evaluate the quality of these predictions, we compare them across all initializations. Our experiments demonstrate high performance with 99.86% Levenshtein accuracy, 99.82% perfect accuracy, and 99.50% digit accuracy (for further details, please refer (Chambers et al., 2019)), indicating strong robustness in the model's predictions.

**What mechanisms or procedures were used to collect the data (e.g., hardware apparatuses or sensors, manual human curation, software programs, software APIs)? How were these mechanisms or procedures validated?** The data is provided to us from Bashpole Software, Inc..

**Did you collect the data from the individuals in question directly, or obtain it via third parties or**

other sources (e.g., websites)? Over what time-frame was the data collected? Does this time-frame match the creation timeframe of the data associated with the instances (e.g., recent crawl of old news articles)? If not, please describe the timeframe in which the data associated with the instances was created. The data collected in this research is scraped from online posted ads between December 2015 and April 2016 on the Backpage Escort Markets.

Were the individuals in question notified about the data collection? If so, please describe (or show with screenshots or other information) how notice was provided, and provide a link or other access point to, or otherwise reproduce, the exact language of the notification itself. No.

Did the individuals in question consent to the collection and use of their data? If so, please describe (or show with screenshots or other information) how consent was requested and provided, and provide a link or other access point to, or otherwise reproduce, the exact language to which the individuals consented. NA

If consent was obtained, were the consenting individuals provided with a mechanism to revoke their consent in the future or for certain uses? If so, please provide a description, as well as a link or other access point to the mechanism (if appropriate). NA

### A.4.4 Preprocessing/cleaning/labeling

Was any preprocessing/cleaning/labeling of the data done (e.g., discretization or bucketing, tokenization, part-of-speech tagging, SIFT feature extraction, removal of instances, processing of missing values)? If so, please provide a description. If not, you may skip the remaining questions in this section. To prioritize privacy and minimize the potential for misuse, we implemented measures to protect sensitive information within the ad descriptions in our dataset. This involved masking various elements, such as phone numbers, email addresses, age details, post IDs, dates, and links mentioned in the ads. By applying these safeguards, we aim to maintain the confidentiality of individuals involved while still enabling valuable research and analysis.

Was the "raw" data saved in addition to the preprocessed/cleaned/labeled data (e.g., to support unanticipated future uses)? If so, please provide a link or other access point to the "raw" data. No.

Is the software that was used to preprocess/clean/label the data available? If so, please provide a link or other access point. No.

### A.4.5 Uses

Has the dataset been used for any tasks already? If so, please provide a description. This research uses the IDTraffickers dataset to establish authorship identification and verification benchmarks.

What (other) tasks could the dataset be used for? None.

Is there anything about the composition of the dataset or the way it was collected and preprocessed/cleaned/labeled that might impact future uses? For example, is there anything that a dataset consumer might need to know to avoid uses that could result in unfair treatment of individuals or groups (e.g., stereotyping, quality of service issues) or other risks or harms (e.g., legal risks, financial harms)? If so, please provide a description. Is there anything a dataset consumer could do to mitigate these risks or harms? Despite our best efforts to mask sensitive information, it is important to note that the dataset still contains certain details such as escort pseudonyms, posted locations, ethnicity, and sexual preferences. Although the likelihood of this information being used to harm individuals is low, we strongly discourage any unethical research or commercial applications that could potentially lead to the identification of escort individuals, either directly or indirectly. Our intention is to prioritize the privacy and well-being of all individuals involved in the dataset.

### A.4.6 Distribution

Will the dataset be distributed to third parties outside of the entity (e.g., company, institution, organization) on behalf of which the dataset was created? If so, please provide a description. Yes, we intend to make our dataset available through the Dataverse data repository to mitigate any potentially illegal or fraudulent use of our dataset. The access to the data will be subject to specific conditions, including the requirement for researchers to sign a non-disclosure agreement (NDA) and data protection agreements. These agreements will prohibit sharing the data and its use for unethical commercial purposes.

**How will the dataset will be distributed (e.g., tar-ball on website, API, GitHub)? Does the dataset have a digital object identifier (DOI)?** Yes

**When will the dataset be distributed?** We will be releasing the dataset after the final decision from the EMNLP committee, along with the camera-ready version.

**Have any third parties imposed IP-based or other restrictions on the data associated with the instances? If so, please describe these restrictions, and provide a link or other access point to, or otherwise reproduce, any relevant licensing terms, as well as any fees associated with these restrictions.** No

### A.4.7 Maintenance

**Will the dataset be updated (e.g., to correct labeling errors, add new instances, delete instances)? If so, please describe how often, by whom, and how updates will be communicated to dataset consumers (e.g., mailing list, GitHub)?** As our research progresses, we are actively exploring NLP-based entity extraction techniques to improve the privacy of individuals in the dataset. Specifically, we are investigating methods to effectively mask escort pseudonyms, posted locations, and ethnicity. Our goal is to develop reliable approaches that ensure the anonymity and confidentiality of individuals involved. To keep the research community informed, we will share our progress and findings through research publications and provide updates to the data description on the Dataverse portal.

**If others want to extend/augment/build on/contribute to the dataset, is there a mechanism for them to do so? If so, please provide a description. Will these contributions be validated/verified? If so, please describe how. If not, why not? Is there a process for communicating/distributing these contributions to dataset consumers? If so, please provide a description.** We encourage researchers to actively contribute to and expand our dataset by extending, augmenting, and building upon it. However, in order to prioritize the safety and well-being of the individuals included in the dataset, we have made the decision to restrict the sharing rights of the dataset. Nevertheless, we warmly welcome fellow researchers to collaborate with us on further advancements related to the dataset. While sharing the dataset itself is not permitted, we will acknowledge and appreciate the contributions of researchers and support their continued research endeavors.

