# OpenReview forum: "IDTraffickers: An Authorship Attribution Dataset to link and connect Potential Human-Trafficking Operations on Text Escort Advertisements"
_EMNLP/2023/Conference — EMNLP 2023 Main_

### Official Review · Reviewer_iA59 · 2023-08-04

**Soundness:** 3

**Ethical Concerns:**

Yes

**Excitement:**

4: Strong: This paper deepens the understanding of some phenomenon or lowers the barriers to an existing research direction.

**Justification For Ethical Concerns:**

Given the dataset IDTraffickers is about Human Trafficking, there needs to be a concerted effort to ensure that Personally Identifiable Information (PII or  P2I) is purged to protect victim identity. The paper describes in adequate detail the filtering steps conducted *by the authors* (lines 235 to 249) but an external review might be useful and necessary.



**Missing References:**

None come to mind.

**Paper Topic And Main Contributions:**

The paper contributes a novel and sizable dataset IDTraffickers to combat human trafficking -- by the methods, I'd say focused in the US -- by highlighting a key challenge: conventional methods require a very hands-on approach on a continuous stream of large-scale data, making it expensive. With LLM/LLM-with-memory models trained on the dataset, a first pass through the large amounts of data can be delegated to the model, and human intervention can be applied as a more fine-grained analysis.

Authors demonstrate relevant data statistics (frequency histograms and POS distributions) including one creative figure captioned "wikifiability", indicating that the scraped dataset is substantial, diverse and grounded. And suggest two tasks -- Authorship Identification and Authorship Verification. Tasks are addressed in distinct and rational setups: Identification is conducted via a standard LLM + Classifier setup; in Verification, the authors choose to use a vector store -- somewhat serving as a recommendation engine.

All-in-all, the combination of dataset and initial model performance baselines appear to suggest that the contribution paves a novel way of addressing human trafficking problems when considering the datasets in real-time and in bulk.


**Questions For The Authors:**

In this case, I would like the authors to comment on the points listed in reasons to reject.

**Reasons To Accept:**

Reasons to accept listed below:

A: The paper is highly relevant and discusses a challenging real-world issue: human trafficking. To address the challenge, the authors propose a new and exciting dataset IDTraffickers that is large enough to fine-tune LLMs. In this way, we can leverage the power of LLMs and vector stores to address a societal issue at scale.

B: The reported statistics are relevant and useful to help demonstrate the quality of the dataset. While I still have concerns about the data itself, I thought the statistics were necessary and near-sufficient.

C: The design of the two tasks and the classification and recall methods to address them -- fine-tuning LLMs and applying vector stores for retriever setups -- is refreshing and cutting-edge.

**Reasons To Reject:**

I can identify a few weaknesses listed below:

A: Drift: Is there stability in the vendors, themes, locations and style of the scraped data points across time, and how much are the trained models susceptible to such drastic changes.

B: Sensitivity: There are several places in the main text -- including an extended discussion in the Appendix -- about data privacy and security. How reliable are these PII-filtering techniques in this dataset? [See flag for Ethical concerns]

C: Correlations between label (vendor) and text structure: I'd like to see a few examples of text *as they correlate to the vendors*. I am curious whether the trained LLM models are learning a proxy function of the vendor and not really using the entire context. The classification performance looks a little too promising...


**Reproducibility:**

4: Could mostly reproduce the results, but there may be some variation because of sample variance or minor variations in their interpretation of the protocol or method.

**Reviewer Confidence:**

3: Pretty sure, but there's a chance I missed something. Although I have a good feel for this area in general, I did not carefully check the paper's details, e.g., the math, experimental design, or novelty.

**Typos Grammar Style And Presentation Improvements:**

I might have missed a few. Kindly note:

A: line 80 - *none* analyzed

B: lines 219, 220, 222 - (citations not in brackets)

C. line 93 - United State: "s" missing

In general, a speling and grammar check might be useful.

D: line 1 - "violating their fundamental human right" - sounds a little obvious.

---

> ### Author Rebuttal · Authors · 2023-08-29
>
> We thank the reviewer for their insightful feedback and for acknowledging our work as relevant, challenging, exciting, refreshing, and cutting-edge. We encourage the reviewer to find answers to their concerns below, and given an opportunity, we would like to integrate our responses to the comments in the camera-ready version of the manuscript.
>
> **A: Drift** The absence of ground truth poses challenges in collecting ads from authors spanning multiple years to comment on the stylometric drift. Our data collection specifically covers escort advertisements between December 2015 and April 2016, during which we observed no significant shifts in authors' writing styles. Moreover, given the brevity of this period, it's challenging for us to ascertain the stability of escort vendors. Notably, establishing the business age for potential human trafficking vendors is intricate, mainly because not all such vendors are apprehended by law enforcement. However, research indicates that more organized traffickers exploit victims extensively over extended timeframes ([2]).
>
> Indeed, we see a drift among vendor locations within our dataset. The extent of this drift across different American geographies is outlined in the following table:
>
> | Geographic Pair | Number of Common Vendors |
> |-----------------|-------------------------|
> | East - West     | 75                      |
> | East - North    | 6                       |
> | East - South    | 107                     |
> | East - Central  | 284                     |
> | West - North    | 7                       |
> | West - South    | 53                      |
> | West - Central  | 34                      |
> | North - South   | 7                       |
> | North - Central | 8                       |
> | South - Central | 207                     |
>
> _Our analysis further reveals that 7 vendors post their advertisements across all 41 states of the United States from which our dataset was collected._
>
> Finally, our methodology employs text ads to identify and establish connections among vendors associated with online escort ads. While our study's principles can be theoretically generalized to any online active escort website, the practical implications may vary based on certain factors, including disparities in writing styles, use of chat-GPT-like automated systems, the presence of different languages, and data noise. Additionally, gathering ads from authors spanning considerable time intervals is challenging. In essence, different people write differently, and since the authorship task relies on identifying individual writing styles and perpetrators of human trafficking, it spans all racial, ethnic, and gender demographics ([2, 3]); generalizing such approaches to vendors' existing years apart is not straightforward. In such instances, we recommend our readers improve the trained model through further fine-tuning, leveraging transfer learning techniques from the IDTraffickers dataset, and applying these to their specific data context.
>
> **B: Sensitivity** As outlined in lines 235-249, 626-655, and detailed in Appendix section A.2, we have implemented extensive measures to safeguard individual identities. These precautions include masking personal information, including phone numbers, email addresses, age details, post IDs, dates, and links. This masking is executed to prevent any possibility of reverse-engineering, thus mitigating the potential for misuse of our dataset.
>
> Unlike other encryption techniques, our masking process eliminates personal information. For instance, an email address like xzy@gmail.com is transformed into <EMAIL_ID_23>, a phone number such as +00 00 0000 000 becomes +NN NN NNNN NNN, and a link like www.google.com is rendered as <LINK> within our dataset. However, due to substantial noise in our dataset, we could not remove the first names of escorts utilized in the advertisements. Nevertheless, considering prior research indicating that escorts frequently use pseudonyms in their ads ([4, 5]), combined with the absence of public records for these ads post the 2018 seizure of Backpage Escort Markets, we assess it as improbable that individuals could exploit the personal data in our ads to cause harm.
>
> Furthermore, given the legitimate interest of our research, our masking methodology aligns with the spirit of the General Data Protection Regulation (GDPR) article 6, the lawfulness of processing ([6]), which acknowledges the importance of balancing privacy concerns with societal benefits. In this context, our research contributes to the betterment of society by facilitating studies aimed at addressing critical issues such as human trafficking and enhancing law enforcement efforts.
>
> Finally, we intend to grant access to the dataset via the Dataverse data repository portal. However, this access is subject to stringent restrictions. Researchers seeking access must undergo an application process, necessitating approvals and assurances. These assurances encompass non-disclosure and data protection agreements, ensuring responsible and secure use and release of the data.
>
> **C: Correlations** We thank the reviewer for pointing this out. Our analysis indicates that our trained classifier employs both mechanisms to distinguish between various vendors' writing styles effectively. Example A illustrates a scenario where our trained model has acquired a proxy function of a particular vendor's writing style. On the other hand, Example B demonstrates an instance where the model has developed the ability to differentiate between vendors by leveraging contextual cues. Please note that both examples are drawn from the true positive predictions within the test dataset.
>
> **_Example A:_** The examples below illustrate the model's inclination to formulate predictions using proxy functions instead of contextual cues. Our explanations clarify that the model assigns greater attribution to tokens presented in uppercase format. This leads us to believe that our trained model associates sequences of consecutive uppercase letters as a proxy function for representing advertisements from Vendor 5801.
>
> **_A1._** COME?? AND ENJOY THE SWEET TOUCH VERY BEAUTIFUL ??SENSUAL LATINAS GIRLS SUPER AMICABLE GOOD TREATE VERY SWEET NICE?? LADIES GOOD MASAJE ??SOF HAND GOOD SERVICE NO RUSH?? WE?? WAITING FOR?? YOU COME PAPI SENSUAL ??HOT NENAS BELLAS ??LATINAS WE PRIORATY?? IS FOR YOU EXCELENTE ??SERVICE ??FOR MORE INF COME WE LOCATION NNNN FLORENCE AVE TELL NNN.NNN.NNNN.& NNN.NNN.NNNN Posters age: <AGE> • Location: BELL HUNTINGTON PARK TELL NNN.NNN.NNNN ?, City of Long Beach, Downtown, Long Beach, Los Angeles, San Gabriel [SEP] <LINK> [SEP] SENSUAL curves EXÓTICAS?? Beautiful Ladies Sexy pretty ??GIRLS WAITING FOR YOU ??sweet Nice Touch?? - Los Angeles escorts - <LINK>
>
> **_A2._** MASSAGE TERAPEUTA SOF TOUCH ALL BODY RELAX SEXY CURVES EXOTIC SENSUAL HOT NENAS BELLAS LATINAS GIRLS SUPER AMICABLE AND SWEET COME BABY WE WAITING FOR YOU GOOD SERVICE NO RUSH WE PRIORITY IS FOR YOUR BEST TREATED WELCOME PAPI ENJOY THE SWEET TOUCH PRETTY GIRLS NICE PLACE AND PRIVATE ROOM BEAUTIFUL LADIES FOR MORE INF COME WE LOCATION NNNN FLORENCE AVE TELL (NNN)NNN.NNNN.& (NNN)NNN.NNNN Posters age: <AGE> • Location: BELL HUNTINGTON PARK TELL NNN.NNN.NNNN?, Downtown, Inglewood / Hawthorne [SEP] <LINK> [SEP] ??SWEET pretty ?? Latinas good ??Services ?? No rushed ?? N SEXY??Staff??Sof ??TOUCH ??good MASAJE?? - Los Angeles escorts - <LINK>
>
> **_A3._** ??PATITE ??SENSUAL??PRETTY?? NENAS LATINAS ??GIRLS COME ??AND ENJOY ????THE?? SWEET ??GOOD ??MASAJE ??EVERYTHING?? YOU NEED?? WE HAVE ????PRETTY ??LADIES ??GOOD ??SERVICE ??NO?? RUSH ??VERY AMICABLE ??GIRLS?? WELCOME ??PAPI WE WAITING ??FOR ??YOU THE PLACE IS?? VERY ??CLEAN ??AND?? PRIVATE ROOM ??FOR MORE ??INF COME ??WE LOCATION NNNN FLORENCE AVE ?? TEL ??NNN.NNN.NNNN & ??NNN.NNN.NNNN Posters age: <AGE> • Location: Downtown, Los Angeles [SEP] <LINK> [SEP] ?? Come On ??over Enjoy??SEDUCTIVE Beautiful ??Latinas?? waiting for ??You COME for good ?? MASAJE?? - Los Angeles escorts - <LINK>
>
> **_Example B:_** The examples below illustrate the model's inclination to establish predictions relying on contextual cues rather than solely depending on a proxy function. While we lack a definitive means of confirming the model's focus on the advertisement context, our explanations highlight pronounced inconsistencies in word attributions for similar tokens across these advertisements. Consequently, our assessment leads us to believe that these predictions are more likely a result of the contextual cues present within the advertisements of Vendor 390.
>
> **_B1._** Yes you read it right lesbians....but we love to show off and spoil a hard working man!!!we are Bbw so please do expect a thin girl. Just be polite and respectful and you will see two girls set ablaze!!! TEXT-NNN-NNN-NNNN S.S.-𝑁𝑁,𝐻.𝐻.−��,�.�.−NNN, Hr.-$NNN... Remember rates for time no illegal acts!!! Text now we are hosting in Lancaster east of downtown Lancaster!!! NNN-NNN-NNNN Posters age: <AGE> • Location: Harrisburg [SEP] <LINK> [SEP] Lesbians looking for a nice guy! - Lancaster escorts - <LINK>
>
> **_B2._** IN CALLS IN LANCASTER ONLY!!!Lancasters luscious lesbian BBWs are looking for a nice ,polite,and generous, gentleman to show a good time !!!?? If this sounds like a good plan to you TEXT US QUICK!!!?? Please?? no women, no couples,no drugs/alcohol, no explicit/inappropriate talk, or multiple men.?? Please be respectful. TEXT-NNN-NNN-NNNN S.S.-??NN?? H.H.-??NNN?? Hr.-??NNN?? Bonus= If you read to the end text the word "candy" to us to get a special surprise!!!?? Posters age: <AGE> • Location: Lancaster [SEP] <LINK> [SEP] Lancasters luscious lesbians!!! - Harrisburg escorts - <LINK>
>
> **_B3._** These are N BBWs You wanna make time for? A very real couple for N years? Sexy??,funny??,intelligent ??,and we are N Lesbians know how to show a guy a good time? We are in Lancaster close to Outlets. In calls only? ??? Please no women,no couples,no drugs/alcohol,no explicit comments,and no multiple guys???!!! PLEASE BE POLITE AND RESPECTFUL!!! ??NN-S.S.?? ??NNN-H.H.?? ??NNN-Hr.?? PLEASE TEXT ONLY-NNN-NNN-NNNN?? Posters age: <AGE> • Location: Lancaster Lincoln highway [SEP] <LINK> [SEP] Too hot for games?! Come join N
>
> **Typos Grammar Style And Presentation Improvements::** We agree with the reviewer's comment. We will ensure to rectify it in our camera-ready version.
>
> **References**
>
> 1. BACKPAGE.COM'S KNOWING FACILITATION OF ONLINE SEX TRAFFICKING - https://www.govinfo.gov/content/pkg/CHRG-115shrg24401/html/CHRG-115shrg24401.htm
> 2. Global Report on Trafficking in Persons 2022 - https://www.unodc.org/documents/data-and-analysis/glotip/2022/GLOTiP_2022_web.pdf
> 3. Myths, Facts, and Statistics - https://polarisproject.org/myths-facts-and-statistics/
> 4. Comparing manual and computational approaches to theme identification in online forums: A case study of a sex work special interest community - https://www.sciencedirect.com/science/article/pii/S2590260121000229
> 5. Indicators of Sex Trafficking in Online Escort Ads - https://www.ojp.gov/pdffiles1/nij/grants/305453.pdf
> 6. Article 6 GDPR. Lawfulness of processing - https://gdpr-text.com/read/article-6/

---

### Official Review · Reviewer_buUC · 2023-08-05

**Soundness:** 4

**Excitement:**

4: Strong: This paper deepens the understanding of some phenomenon or lowers the barriers to an existing research direction.

**Paper Topic And Main Contributions:**

Human trafficking (HT) is a societal problem and has online presense in the form of escort ads on certain websites. While the law enforcement agencies are working to tackle the problem, they need tools, and tools and models that need data.
This works presents a dataset and plans to release dataset,

In addition, they also present baseline models to identify(ID) the vendors via authorship verification. They have two tasks: classification and ranking task, where the latter is more generalizable.
The authors attempt to mask some direct clues (such as phone numbers, email ID, etc ) that directly indicate author ID, and instead expect the models to identify cluster of ads that are from the same author.


**Questions For The Authors:**

* Q1) Line 211: What languages are found in the US-only filtered dataset (besides English)?

* Q2) Line 367-369: Surprised to see only 1% improvement (expected better). Wondering if multilingual models (such as mBERT, Roberta-xlmr etc) made any difference.

* What is your take on generalization of styles to authors that maybe many years apart. Are HT vendors short lived-in their business, or do they have a career that lasts over a decade?


**Reasons To Accept:**

* This paper presents a unique (perhaps rare) resource to the NLP/CL community.
* Presents a baseline model, thus accelerating a future development that revise and improve quality of authorship verification models.

**Reasons To Reject:**

Dataset comes from 2015/12 - 2016/04, perhaps a bit outdated regarding what has happened on the Internet.


**Reproducibility:**

3: Could reproduce the results with some difficulty. The settings of parameters are underspecified or subjectively determined; the training/evaluation data are not widely available.

**Reviewer Confidence:**

3: Pretty sure, but there's a chance I missed something. Although I have a good feel for this area in general, I did not carefully check the paper's details, e.g., the math, experimental design, or novelty.

---

> ### Author Rebuttal · Authors · 2023-08-29
>
> We thank the reviewer for their insightful feedback and for acknowledging the uniqueness of our work. We encourage the reviewer to find answers to their concerns below, and given an opportunity, we would like to integrate our responses to the comments in the camera-ready version of the manuscript.
>
> **Response to the rejection concern:** We acknowledge the reviewer's concern regarding our dataset's age and regret not addressing it in our limitations section. We intend to address this oversight in the final version of our manuscript. Despite the presence of several active online escort platforms like Craiglist, YesBackpage, and Classifiedads, we focus our research on Backpage ads spanning December 2015 to April 2016. This choice stems from documented human trafficking (HT) activities on the Backpage website ([1]). We opted not to incorporate other active websites due to the absence of HT information in data from other escort websites. Please note that our research does not aim to track or assist currently ongoing human trafficking investigations. Instead, we aim to showcase the potential of authorship attribution techniques to aid law enforcement agencies in resource prioritization. Given this research objective, the age of the dataset holds smaller significance.
>
> **Q1:** Our analysis shows that around 99.32 % of the vocabulary in our dataset is from the English language. Complete details about the presence of all the languages are as follows:
>
> | **Language** | **Frequency** |
> |----------|----------------------|
> | ENGLISH  | 99.3200 %            |
> | SPANISH  | 0.4378 %             |
> | CHINESE  | 0.1382 %             |
> | JAPANESE | 0.0207 %             |
> | SOTHO    | 0.0196 %             |
> | FRENCH   | 0.0127 %             |
> | BOKMAL   | 0.0103 %             |
> | DUTCH    | 0.0058 %             |
> | TSWANA   | 0.0058 %             |
> | NYNORSK  | 0.0046 %             |
> | SWEDISH  | 0.0046 %             |
> | LATIN    | 0.0035 %             |
> | BASQUE   | 0.0034 %             |
> | SWAHILI  | 0.0023 %             |
> | MALAY    | 0.0023 %             |
> | XHOSA    | 0.0023 %             |
> | WELSH    | 0.0023 %             |
>
> These statistics are obtained using the lingua language detection (https://github.com/pemistahl/lingua-py) Python library.
>
> **Q2:** As indicated above, the vast majority of vocabulary in our dataset (over 99%) belongs to the English language. Consequently, we did not explore the use of multilingual models in our research. Given that only a small fraction of our dataset's vocabulary lies outside English, we anticipate employing multilingual models would not significantly improve performance.
>
> As detailed in lines 365-366, the language model trained on our dataset achieves a test perplexity of 6.22. However, this relatively high perplexity is expected due to noise within our dataset. Escort vendors intentionally introduce random noise into their advertisements to evade detection by automated systems. Consequently, the trained language model faces challenges in predicting whether the next token in the sequence is a word, a symbol, an emoji, or a series of whitespaces. We hypothesize that the minor <1% improvement in the performance of the trained classifier can be attributed, in part, to this random noise effect. This discussion can be found in lines 366-369 of our manuscript.
>
> **Q3:** Due to the absence of ground truth, collecting ads from authors spanning multiple years presents challenges. Determining the business age for potential human trafficking vendors is difficult, especially considering that law enforcement can not apprehend all such vendors. However, it is established that more organized traffickers exploit victims extensively over extended timeframes ([2]).
>
> In essence, different people write differently, and since the authorship task relies on identifying individual writing styles and perpetrators of human trafficking spans all racial, ethnic, and gender demographics ([2, 3]); generalizing such approaches to vendors' existing years apart is not straightforward. Our methodology employs text ads to identify and establish connections among vendors associated with online escort ads. While the fundamental principles of our research can be theoretically extended to any active online escort website, the generalization capability of our approaches depends upon variations in writing styles, the use of chat-GPT-like automated systems, the presence of different languages, and data noise. In such instances, we recommend our readers improve the trained model through further fine-tuning, leveraging transfer learning techniques from the IDTraffickers dataset, and applying these to their specific data context.
>
> **Reproducibility:** We plan to attach the GitHub repository to our code in our camera-ready version for better reproducibility.
>
> **References:**
>
> 1. BACKPAGE.COM'S KNOWING FACILITATION OF ONLINE SEX TRAFFICKING - https://www.govinfo.gov/content/pkg/CHRG-115shrg24401/html/CHRG-115shrg24401.htm
> 2. Global Report on Trafficking in Persons 2022 - https://www.unodc.org/documents/data-and-analysis/glotip/2022/GLOTiP_2022_web.pdf
> 3. Myths, Facts, and Statistics - https://polarisproject.org/myths-facts-and-statistics/

---

### Official Review · Reviewer_fd99 · 2023-08-05

**Soundness:** 4

**Excitement:**

4: Strong: This paper deepens the understanding of some phenomenon or lowers the barriers to an existing research direction.

**Paper Topic And Main Contributions:**

This paper presents an approach based on authorship attribution to identify potential human trafficking vendors on an online escort market currently seized (Backpage) and achieve a macro-F1 score of 0.87. Then they perform an authorship verification based on the extracted style representation with a mean r-presicion score of 0.89.

**Questions For The Authors:**

Q-A: What do you mean by "closed-set/open-set verification/classification/identification"? Please clarify where you used each.
Q-B: Is Figure 2 (A) the number of tokens or sentences? Based on the numbers, I guess it is tokens, but the label (SENT-LEN) is misleading.
Q-C: Apparently, Backpage has been seized since 2018! But other websites like Craigslist (as you mentioned) are still active, and I assume your proposed approach could be beneficial in identifying HT on these other websites too. Can you give more examples, and briefly discuss how your approach can be suited for a few other similar prominent websites?
Q-D: It appears from the Appendix that you have not collected any media/images from these advertisements. But, have you performed any manual analysis to see if there are duplicates that could be used for improving the authorship attribution task? (e.g., if the ads have used similar/same images, they might be from the same vendor or fake). At least you may want to briefly discuss this as a future research direction.


**Reasons To Accept:**

A1: Human trafficking is an important and sensitive matter, especially in the escort market and adult services industry. Identifying such advertisements and vendors can help law enforcement identify potential human and sex trafficking incidents faster.
A2: The authors collected a relatively large corpus of escort advertisements from the Backpage website before it was seized. The corpus contains over 87k ads from 14 different states.
A3: The authors evaluate their proposed authorship attribution with other authorship attribution tasks as well as through an authorship verification of the present dataset


**Reasons To Reject:**

R1: While this work aims to propose an approach based on authorship attribution for the identification of potential human trafficking vendors, in the end, it only helps identify ads posted by different vendors. This is because the authors do not have ground truth for vendors with the real risk of human trafficking and cannot claim their performance in detecting such vendors. However, I assume this is not unusual, and gaining access to such information would be difficult, if possible at all, for every vendor. And still, the proposed approach is domain-specific and takes into account the unique characteristics of an escort market website which is slightly different from any other authorship attribution task and could help in the detection of potentially suspicious vendors and advertisements.

**Reproducibility:**

4: Could mostly reproduce the results, but there may be some variation because of sample variance or minor variations in their interpretation of the protocol or method.

**Reviewer Confidence:**

4: Quite sure. I tried to check the important points carefully. It's unlikely, though conceivable, that I missed something that should affect my ratings.

**Typos Grammar Style And Presentation Improvements:**

T1: "authorship techniques" and "authorship approaches" have been used in the paper. But they are not always correct. The authorship approaches are not discussed in this paper. Instead, this paper discusses authorship "attribution/verification" techniques/approaches and maybe authorship-based approaches for detecting HT.
T2: Since the reader reads each page from top to bottom, it would look/feel more natural if the flow of figures (Figure 1, for ex.) is from top to bottom and the starting point is clearer (especially in Figure 1 (i)).
T3: The related work section has focused on authorship attribution, but one of the major tasks in this paper is authorship verification. That also needs to be discussed with enough emphasis in this section, even though some of these references and methods might be shared among attribution and verification.
T4: You should place the footnote marks after the punctuation (line 201, for ex.)

---

> ### Author Rebuttal · Authors · 2023-08-29
>
> We thank the reviewer for their insightful feedback and for acknowledging the importance of our work. We encourage the reviewer to find answers to their concerns below, and given an opportunity, we would like to integrate our responses to the comments in the camera-ready version of the manuscript.
>
> **R1:** We acknowledge the reviewer's valid concern regarding establishing definitive ground truth for escort advertisements (ads) associated with human trafficking (HT). This difficulty is inherent to measuring illegal activities: It is impossible to find undetected crime. Even data based on arrest records does not include undetected ads. Because we wanted to ensure that we use data that included HT ads, we analyzed Backpage escort website ads, as prior investigations have revealed instances of HT ads on the Backpage escort website ([1]). Furthermore, it is important to clarify that our primary objective is not to pinpoint HT ads directly. Instead, and as indicated in lines 656-665 of the manuscript, we aim to offer a tool that supports law enforcement in prioritizing their resources effectively by finding potential HT ads.
>
> **QA:** The closed-set authorship identification setting utilizes a classification task to analyze the writing styles of input ads and assign them to one of the predefined vendor classes ([2, 3]). Essentially, the objective is to classify textual ads into predetermined vendor labels (say vendor 1 - vendor 100) available during the training. Figure 1 (ii), Section 4.1, and Section 5.1 demonstrate this task's setup and the results. In contrast, the open-set authorship verification scenario utilizes a ranking task to enables a similarity analysis between vendor ads from established and _potential future vendors_ (say vendor 501). Importantly, this includes vendors that were not encountered during the training phase. Figure 1 (iii), Section 4.2, and Section 5.2 demonstrate the setup and the results associated with this task.
>
> **QB:** We thank the reviewer for pointing this out and apologize for the confusion. The values on the y-axis of Figure 2 (A) represent the number of tokens per ad. We plan to correct the label on the x-axis in the camera-ready version to avoid confusion.
>
> **QC:** The reason for not focusing on other active websites is related to the unavailability of comparable HT information in the data on those websites.
>
> Regarding using our approach for other websites, our methodology employs text ads to identify and establish connections among vendors associated with online escort ads. While the fundamental principles of our research can be theoretically extended to any active online escort website, the generalization capability of our approaches depends upon variations in writing styles, the use of chat-GPT-like automated systems, the presence of different languages, and data noise. In such instances, we recommend our readers improve the trained model through further fine-tuning, leveraging transfer learning techniques from the IDTraffickers dataset, and applying these to their specific data context.
>
> **QD:** We concur with the reviewer's observation regarding our dataset's potential use of images as soft labels. This aspect holds promise and is currently a focal point of our research efforts. We intend to present our findings in upcoming research. That said, we plan to incorporate this aspect into the discussion section of the finalized camera-ready version.
>
> **T1, T2, T4:** We agree with the reviewer's comment. We will ensure to rectify it in our camera-ready version.
>
> **T3:** We agree with the reviewer's comment here. We plan to include the literature for the authorship verification task in our camera-ready version.
>
> **References**
>
> 1. BACKPAGE.COM'S KNOWING FACILITATION OF ONLINE SEX TRAFFICKING - https://www.govinfo.gov/content/pkg/CHRG-115shrg24401/html/CHRG-115shrg24401.htm
> 2. Open Set Text Classification using Convolutional Neural Networks - https://aclanthology.org/W17-7557.pdf
> 3. Open-Set Recognition: a Good Closed-Set Classifier is All You Need? - https://arxiv.org/abs/2110.06207

---

### Meta-Review · Area_Chair_zAhw · 2023-09-22

**Recommendation:** 5

**Metareview:**

The study introduced a dataset aimed at potential human trafficking operations, comprising 87,595 text ads and 5,244 vendor labels. This addresses a significant social issue, and reviewers agree that the dataset will be a valuable asset to the research community.
Various important points were brought up by the reviewers that should be considered to enhance the paper. For instance, Reviewer buUC highlighted i) the time frame of the dataset, ii) multilingual models - providing some justification through experimental results would be more beneficial than mere speculation.

Reviewer iA59 pointed out ethical concerns, which the authors addressed in the rebuttal. Such discussion should be elaborated upon in the paper for better clarity.

---

### Decision · Program_Chairs · 2023-10-07

**Decision:**

Accept-Main

**Comment:**

The study introduced a dataset aimed at potential human trafficking operations, comprising 87,595 text ads and 5,244 vendor labels. This addresses a significant social issue, and reviewers agree that the dataset will be a valuable asset to the research community.
Various important points were brought up by the reviewers that should be considered to enhance the paper. For instance, Reviewer buUC highlighted i) the time frame of the dataset, ii) multilingual models - providing some justification through experimental results would be more beneficial than mere speculation.

Reviewer iA59 pointed out ethical concerns, which the authors addressed in the rebuttal. Such discussion should be elaborated upon in the paper for better clarity.